# Health impact assessment and cost–benefit analysis: Exploring complementarities of methods to assess the impacts of regulations on food consumption

Constanza De Matteu Monteiro[1]*, Rodney Feliciano[2], Jeanne-Marie Membré[2], Sara Monteiro Pires[1], Sofie Theresa Thomsen[1], Stéphan Marette[3]

1 National Food Institute, Technical University of Denmark, Kongens Lyngby, Denmark, 2 Oniris VetAgroBio, INRAE, Secalim, Nantes, France, 3 Université Paris-Saclay, INRAE, AgroParisTech, Paris-Saclay Applied Economics, Palaiseau, France

* cdmmo@food.dtu.dk

## Abstract

This paper proposes an interdisciplinary framework that combines different methodologies to measure the risks and benefits related to dietary patterns and to assess the impact of possible regulations influencing food consumption. First, we briefly review the relevant methodologies within the field of health-economics. Based on the gaps identified in this review, we proposed a framework allowing an integrated application of quantitative health impact assessment (HIA) using disability–adjusted life year (DALY) and cost–benefit analysis (CBA) calibrated with measures capturing consumers preferences, such as willingness to pay (WTP) and purchase intent (PI). We applied the framework to a case study focusing on the lentil market to test the feasibility of the methodology and we discuss options for model extensions and their limits. We estimated annual DALYs attributable to an increase in the consumption of lentils (scenarios with or without food substitutions), and measured welfare variations and the impact of regulatory instruments, such as subsidies and taxes. Results showed that a hypothetical increase of 30% or more in the consumption of lentils would not be socially accepted as an immediate market reaction, even if lentils are considered a food substitute to reduce the consumption of unprocessed red meat. Our framework is useful to streamline regulatory interventions as it supports the evaluation of impacts of various regulatory instruments, relevant for policies governing consumers' awareness or impacting consumers' choice via incentives. Furthermore, it provides a starting point for further interdisciplinary discussions on holistic assessment supporting food systems change.

**Data availability statement:** All relevant data are within the manuscript and its Supporting Information files.

**Funding:** Holifood project (https://holifood-project.eu/) financed by the European Union's Horizon programme for research and innovation (Grant Agreement No. 101059813 352).

**Competing interests:** The authors have declared that no competing interests exist.

## 1. Introduction

Regulatory interventions for addressing health and environmental impacts of food systems abound globally. These interventions are often enforced to limit various types of risks, externalities and/or information asymmetries, which are pervasive in unregulated markets [1,2]. However, deciding whether to intervene is challenging. It requires the assessment of when a public health issue should be prioritized, and the evaluation of impacts of different regulatory instruments on both health and economic outcomes. In the context of diets and food systems, intervention strategies through regulatory instruments can aim to (i) provide information to consumers through advertising campaigns and/or labelling, (ii) influence consumer consumption habits through the implementation of fiscal policies such as subsidies and/or taxes, or (iii) modify the regulation of practices in terms of quality standards in food production and manufacturing.

There are several existing methodologies available to support different stages of regulatory decision-making processes. Considering the methodological limitations of each approach and the complexity of covering all the dimensions pertinent to food regulation, it may also be challenging for regulatory bodies to identify which method to use and to comprehensively evaluate outputs across several disciplines [3–5]. For instance, while considerable research efforts have been focused on integrating health and environmental dimensions into assessments that support the development of dietary guidelines more coherent with sustainability goals [6–9], existing approaches still fail to capture the economic and political conditions needed to achieve these optimal dietary targets for food systems' change [10,11].

Driven by the need to understand the possible complementarities of different existing methods for improving currently available decision-support frameworks, the aim of this study was to develop an interdisciplinary approach for realizing health and economic assessments that more efficiently support the evaluation of regulatory instruments impacting consumers' food choices and health.

To achieve this aim, first, we briefly reviewed the numerous methodologies that support the evaluation of trade-offs in risk prioritization (e.g., a broader analysis that goes beyond food safety and nutrition) and regulation [12]. The review served to highlight the aspects currently overlooked by existing approaches and point out the gaps we covered in this study. Second, we developed the theorical framework which combines health impact assessment (HIA) with "disability–adjusted life year" (DALY) and cost–benefit analysis (CBA) approach accounting for "willingness to pay" (WTP). Third, we validated our framework through an empirical application using the lentil market in France as a case-study. The shift to an increased consumption in legumes and other plant-based protein sources are at the core of the necessary transition towards sustainable diets [13]. Lentils were deemed as a relevant case-study due to their nutritional benefits (containing protein, micronutrients, bioactive compounds), food ingredient flexibility and agricultural utility (via nitrogen fixation). Moreover, it is a widely accepted legume in France, and a crop that is expected to continue to have an increased demand in Europe [14–18].

## 1.1 Overview of current methodologies

There are several metrics for quantifying the impact of adverse health effects and for evaluating the reduction in health risks applied in various assessment approaches [12,19,20]. Without being exhaustive, this section presents a short review of these methodologies, with a focus on health and economics. Given the plurality of methods and how they can overlap, we map the main methodologies and the elements they cover in Table 1.

Approaches that quantify adverse health effects can be performed with different purposes, such as to estimate the current impact of diseases given a certain exposure (e.g., burden of disease studies using DALY) or to derive health–based guidance values that ensure the safety of foods for consumers (e.g., risk assessment using risk statistics). In the context of HIA, the purpose can be to evaluate the trade-offs between potential adverse and beneficial effects of foods while defining recommendations (e.g., risk–benefit assessment – RBA), estimating optimal consumption scenarios to maximize health or evaluating the efficacy of interventions (various approaches).

When limited resources need to be prioritized and allocated, international authorities may adopt studies using DALY [21,22]. This composite indicator incorporates both disease occurrence and severity, which allows comparisons across diseases while translating health losses at the population level [23,24]. In RBA, DALY or "quality-adjusted life year" (QALY) are applied as common metrics to integrate adverse and beneficial health effects while comparing reference and alternative scenarios [25]. To date, only a few RBA studies have used QALY, as this metric is most often applied in the context of evaluating the efficacy of policies, such as in cost–utility analysis (CUA) or cost–effectiveness analysis (CEA) [26–28].

Except for "multi-criteria decision analysis" (MCDA), an umbrella term for approaches that rank options involving several interrelated criteria, the contribution of the methods discussed above mainly helps identifying when food regulators should intervene. MCDA is a flexible method that can support both risk ranking and risk prioritization and is recommended

**Table 1.  Overview of methods that quantify impacts and inform decision-making problems in the context of health and economic assessments.**

| Methods for quantification of impacts | Elements | | | | |
|---|---|---|---|---|---|
| | Health and Medical care | Citizens Utility | Monetary value | Market mechanism | Regulatory tools |
| Risk statistics | x | | | | |
| DALY | x | | | | |
| QALY | x | x | | | |
| WTP | | x | x | | |
| VSL | x | | x | | |
| **Methods to inform decision–making problems** | | | | | |
| Burden of disease | x | | | | |
| Cost of illness | x | | x | | |
| Risk assessment | x | | | | |
| Risk–benefit assessment | x | | | | |
| Mathematical optimization | x | | x | | |
| Cost–effectiveness analysis | x | x | x | | x |
| Cost–utility analysis | | x | x | | |
| Cost–benefit analysis | | x | x | x | x |
| Multi-criteria decision analysis | x | | x | | x |

DALY, disability-adjusted life year; QALY, quality-adjusted life year; WTP, willingness to pay; VSL, value of statistical life.

by regulatory authorities such as the FAO and WHO to support structured and transparent decision-making in risk management [29]. However, this approach is generally semiquantitative and does not capture some important elements of the economic dimension (Table 1) [5,12,29–32].

From a different perspective, economic assessments support regulatory processes by answering practical questions such as how food regulators should intervene, providing, for example, information on potential costs of the implementation or reinforcement of regulatory instruments to improve health. The contribution of economic assessments is to incorporate overall welfare (i.e., consumer preferences and the impact of indirect effects from externalities) and account for market effects concerning products when handling supply and demand issues.

From the point of view of the cost of diseases to society, the cost of illness analysis is a common "intermediate approach" between the health and economic fields. It can be considered an extension of the burden of disease methods, where DALY or QALY are used to compute the costs of medical treatment, loss of productivity, and loss of life, linking population summary metrics with production measures such as gross domestic product [19,20]. This approach can sometimes generate debate over the monetization of intangible costs related to health.

Another common measure used to value changes in health and mortality in health economics is WTP. Both the QALY and WTP methods are based on utility theory (a concept postulated in microeconomics that reflects citizens' preferences and explains consumer behaviors) [26,33]. Under the WTP and CBA approaches, the value of statistical life (VSL) corresponds to the monetary value that a person is willing to pay for a mortality risk reduction [34]. The VSL approach is used by a recent tool launched by WHO-Europe for the quantification of economic impacts linked to diets [35]. There are other types of economic approaches, such as the CUA or the CEA, which partially apply economic quantification [3,36]. As health care, food safety and nutritional policies evolve, some authors suggest that CUA or MCDA are likely more suitable to support risk management since the quantification of income and the price effects of regulation needed in CBA can be a resourceful task [37].

Previous contributions have highlighted differences and similarities between the different methods, for instance, DALY versus QALY [26,38], or QALY versus WTP [33], and highlighted important considerations of the underlying assumptions and impact of each method [33,38]. Recent publications have further explored the link between QALY and WTP to understand how consumers' WTP could vary when QALY change [27,39–41], but studies examining the potential complementarities between DALY and WTP have not been identified.

Through this brief review, we identified the need for assessments that incorporate market mechanisms and social welfare to further understand the economic aspects of food systems transformation [10]. Thus, following a food product and demand perspective we developed an approach to interconnect models from HIA and CBA based on similar consumption scenarios. The linkages between the models are indirect since their metrics differ, but bridges are possible since for a given change in DALY due to a shift in consumption, a corresponding welfare measure based on WTP can provided, and vice versa. This novel approach to conduct an "expanded" CBA covers the many elements illustrated in Table 1, integrating with transparency health dimensions, market effects, and regulatory instruments, allowing to simultaneously evaluate the need, feasibility, and efficacy of interventions via regulatory instruments.

## 2. Methods

We developed the theoretical approach interconnecting DALY measures from HIA and WTP for conducting an expanded CBA. The explanation of the overall framework will be supported by simplified models for illustration purposes. The application of the framework to the case study details additional considerations regarding technical adaptations and extensions of the models.

### 2.1 The theorical framework

#### 2.1.1 Disability-adjusted life year (DALY). 
DALY combines disease incidence, duration and severity (i.e., "years lived with disability", YLD), with mortality and life expectancy (i.e., "years of life lost", YLL) (see section 1.1). In HIA, DALY

serves as common metric to estimate the overall health impact in terms of the direct attributable annual health loss of disease(s) given a food-related risk factor, Eq. (1).

$$DALY_{h,q} = \sum_{h} (YLD_h + YLL_h)$$

(1)

Where $h$ is the selected health outcomes associated with a risk or protective factor attributed to a food product at a given consumption quantity, $q$. To capture the difference between exposure scenarios, RBA used as a framework for HIA often reports on the DALY difference, $\Delta DALY_{h,q}$ between a reference (*ref*) and alternative (*alt*) scenario, calculated as follows:

$$\Delta DALY_{h,q} = DALY_{h,alt} - DALY_{h,ref}$$

(2)

A positive $\Delta DALY_{h,q}$ is interpreted as a health loss while a negative difference (averted DALY), implies a health gain [42].

There are different approaches to calculate DALY. In this paper, we link consumption with the dose–response functions derived from epidemiological studies following a top–down approach [43–45]. Thus, we capture the change in incidence and mortality due to a change in exposure by calculating the population impact fraction (PIF) based on the relative risks (RR) of $h$ in each scenario. The PIF is then multiplied by the so called "DALY envelope" (i.e., the total burden of disease in a population) [23,45–47].

**2.1.2 Willingness to pay (WTP).** WTP represents the maximum amount of money that a consumer is willing to pay for one or several units of product(s). In our context, we focus on WTP for one unit of a product that can differ regarding quality or safety, illustrating the consumer's preferences regarding the trade-off between the maximum amount of money to allocate and health. The notation corresponds to WTP for one unit with two different levels of quality (or safety) as possibly elicited by applied microeconomics such as lab experiments. Many studies elicit WTP via lab experiments, surveys or econometric studies, where the WTP per unit of product is measured before ($WTP_1$) and after ($WTP_2$) participants are exposed to information concerning health-related characteristic of a product.

$$WTP_2 - WTP_1$$

(3)

This variation gives important information about the monetary premium given a health characteristic, even if determining a per-unit WTP for a specific product in a lab or survey setting does not represent the real decision context in which consumers go through when purchasing goods in supermarkets. Although this is a potential source of bias in WTP elicitations, getting a price measure from lab experiments (or survey) is valuable for improving our understanding regarding how the demand curve could shift upwards or downwards based on consumers' preferences.

The approach that we developed enables us to go a step further by quantifying the impacts of shifts in consumption quantities based on the HIA models (presented above) and related to the shifts in the demand curve coming from the WTP variations. The model introduced in the next section is based on a simplified Marshallian approach (see Eq. 4 in subsection 2.1.3), allowing demand adjustments related to WTP variations (see Eq. 11 and 12, in subsection 2.2.3). We acknowledge that data from experimental economics can also be integrated following different approaches, i.e., based on a Hicksian surplus [48].

**2.1.3 Cost–benefit analysis (CBA) with WTP.** A simplified partial equilibrium model allowing welfare analysis is presented with a focus on consumers aggregated demand for $q$ unit of a specific product. The characterization of consumers' preferences with (un)awareness regarding some characteristics follows the theorical basis of previous studies [49,50]. These consumers' preferences are represented by the utility function ($U$) in Eq. (4), depending on the quantity consumed of $q$, which can be used to derive and calibrate demand curves. The demand curve is determined by maximizing this utility which is subject to (*s.t.*) the budget constraint, classically represented by the income $R$ in relation to the price $p$.

$$\begin{cases} U(q, w) = aq - \frac{bq^2}{2} + I_f fq - I_e eq + w \\ \\ s.t. \quad R = pq + w \end{cases} \tag{4}$$

Where $q$ is the quantity of the product of interest, $w$ is the numeraire good (i.e., unit of measure by which we can measure all the other units of a basket of options), with the parameters $a$, $b > 0$, allowing to capture the immediate satisfaction from consuming these products. The indicative variable $I_f$ and $I_e$ represents the consumers awareness (i.e., knowledge linked to the information revealed to consumers) at the time of purchase and the parameters $f$ and $e$ represent a per-unit benefit and a per-unit risk related to the product, that will be derived from WTP elicited in the lab experiment. The terms $I_f$ and $I_e$ will equal zero when consumers are ignorant about health effects (i.e., non-internalized in the demand). Conversely, when consumers are perfectly aware about the health characteristic(s), $I_f$ and $I_e$ will equal 1, affecting the demand curve as customers internalize this effect through their food choices. Hence, non-internalized health effects, denoted $NIH$ by consumers, are equal to

$$NIH = (1 - I_f)fq - (1 - I_e)eq \tag{5}$$

NIH is equal to zero when $I_f$, $I_e = 1$, since consumers integrate these effects in the utility.

For the consumer, who will search to maximize its utility subject to the budget constraint, the utility $U(q, w)$ integrating the reformulated budget constraint (with $w = R - pq$) can be rewritten as $U(q, R - pq)$ leading to

$$U(q, p, I_f, I_e) = aq - \frac{bq^2}{2} + I_f fq - I_e eq + R - pq \tag{6}$$

The decision maker accounts for both consumers' utility and NIH, leading to a welfare equal to

$$W(q, p) = U(q, p, I_f, I_e) + NIH = a - \frac{bq^2}{2} + fq - eq + R - pq \tag{7}$$

**2.1.4 Evaluating the need for intervention based on DALY and welfare.** To clarify how changes in consumption can affect the direction of health impacts and welfare, we illustrate how DALY and welfare can be integrated in Fig 1, depicting the utility $U(q, p, I_f, I_e)$ and allowing us to compare this measure with the overall $DALY_{h,q}$. To simplify this theorical demonstration we assume $e = 0$, abstracting from this factor in our graphical representation but maintaining in our notations as it will be relevant in section 4. In both graphs of Fig 1, the quantity of the food product $q$ is represented on the x-axis. The y-axis on graph A represents $U(q, p_1, 0, 0)$ and $W(q, p_1)$ assuming a given price $p_1$, and in graph B it depicts the $DALY_{h,q}$ attributed to the hypothetical consumptions of $q$. Different areas for potential intervention are now visible (areas 1–3), with a clear suggestion of most effective ranges in which regulatory interventions could be targeted (area 1), considering quantified health gains $-\Delta DALY_{h,q}$ while respecting consumers satisfaction based on $U(q, p_1, 0, 0)$, $W(q, p_1)$. The graph also allows for projections on how far a potential nutritional target (represented as $Q_3$) would depart from the equilibrium quantity $Q_1$, given the equilibrium price $p_1$, compared to $Q_2$, the equilibrium quantity when utility is maximized. The distance between $Q_2$ and $Q_3$ is of relevance as it represents the impact of change in consumption in terms of domestic surplus, i.e., the cost for consumers if they were to adopt recommendations that goes beyond their preferences, like drastically changing their current dietary habits.

**2.1.5. Market adjustments.** The maximization of the utility $U(q, p, I_f, I_e)$ with respect to $q$, under the budget constraint with a price $p$ results in the following inverse demand for $p$ and $q$:

$$p(Q, I_f, I_e) = Max[a - bq + I_f f - I_e e, 0] \tag{8}$$

And the following demand

$$Q(p, I_f, I_e) = Max \left[ \frac{a - p + I_f f - I_e e}{b}, 0 \right]$$

(9)

The inverse demand can be represented in Fig 2, with the quantity represented on the x-axis and the price represented on the y-axis with the supply curve. To simplify, let us assume that the supply is perfectly elastic, hence that the equilibrium price is constant $p = p_1$. In addition, we omit the revenue effect (e.g., the wealth of consumers can vary over time).

Fig 2 demonstrates the model mechanism when information is revealed. The representation is also useful to illustrate how the model can be empirically calibrated with WTP measures.

For consumers where $I_f = 0$ the demand slope is represented by $a$, with $f$ representing the per-unit benefit which is not yet internalized, and area A the consumers' domestic surplus (i.e., wealth available for expenditure on other goods). After the revelation of the information $I_f = 1$, area B is internalized, and it is possible to estimate a proxy for an increased demand with $a + f$, where the distance between demands $a$ and $a + f$ is the cost of ignorance (e.g., lack of information of consumers leading to suboptimal decision in terms of food-health trade-off), moving the demand curve vertically, as well as estimate the difference between $Q_1$ and $Q_2$, $Q_2$ being the new projected equilibrium quantity when information is internalized when $p_1$ remains constant. The increase in consumers domestic surplus when the demand is maximized is represented by area D, since $B = C$ when the demand $a + f$. Note that $f$ (or $e$) can be estimated by integrating the WTP variation Eq. (3) as will be explained in the next section. Similarly, $\Delta Q$, representing the consumers' purchase intention, can also be estimated from quantity variations elicited in survey, lab or field experiment.

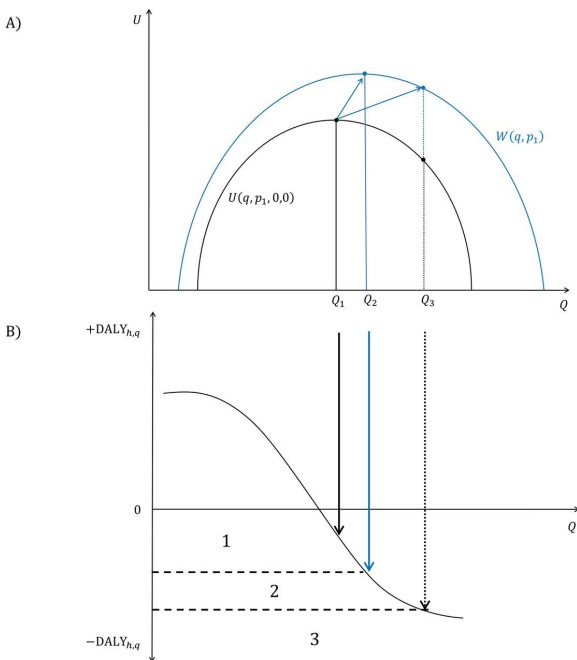

**Fig 1. DALY and welfare.** Note: Simplified theorical representation of welfare and DALY for comparison of utility measures in relation to a hypothetical trend in health gains. Intervention areas are drawn based on the quantity consumed of a food product, $q$. A) Welfare graph: black curve depicts utility when $If = 0$. The blue curve illustrates the welfare. B) DALY graph: Different intervention areas are drawn based on projected $Q$ estimates. Area 3 could be considered as a threshold for non-intervention.

**2.1.6 Impact of regulatory instruments.** Several types of regulatory interventions may influence consumption habits, namely (1) information, (2) fiscal policies, and (3) norms and minimum-quality standards (which are not detailed in this paper). Interventions targeting increasing awareness of consumers through labels, recommendations, and other advertising campaigns, should consider that information reaching consumers is usually imperfect and can lead to under- or over-reaction due to several reasons: information overload, information proliferation, and other competitive factors present in real food purchasing contexts. Thus, it may be expected that interventions only targeting information to consumers would yield increased purchases (and therefore, affecting the demand), but mainly ranging between $a$ and $a + f$ (Fig 2). The mechanisms for fiscal interventions such as subsidy are illustrated in Fig 3.

When information is not chosen as an instrument for the internalization of $f$, decision-makers may alternatively maximize the welfare by indirectly reducing the cost of ignorance via a per-unit subsidy $s$, subtracted to the market price and passed onto consumers. Indeed, with our "simplified" supply side under constant return to scale with perfect competition between producers, the market price is equal to the marginal cost. In this context, the per-unit subsidy (respectively tax) given (respectively imposed) to producers is subtracted (or added) to the market price initially equal to the marginal cost, since the per-unit tax/subsidy is fully passed onto consumers [51]. In this way, $f$ is internalized with ignorant consumers,

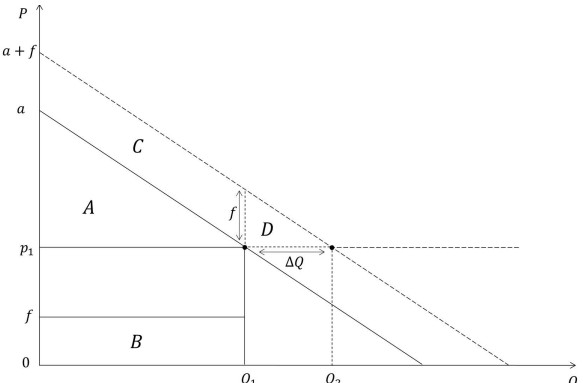

**Fig 2. Market adjustments for impact of information on demand.**

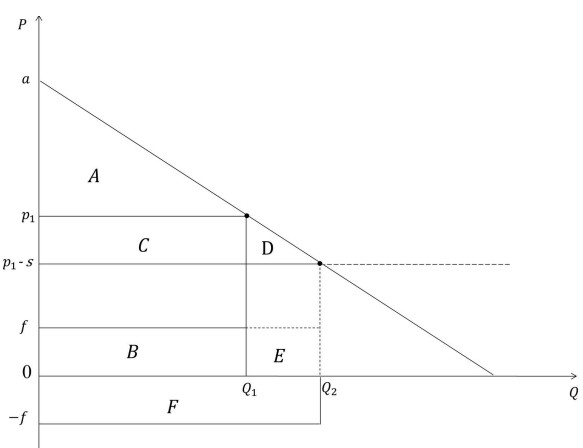

**Fig 3. Intervention mechanisms for fiscal instruments such as subsidies.**

by reducing $p_1$ to $p_1 - s$ and moving $Q_1$ to $Q_2$ with $f = s$. Once again for simplicity we assume that the price variation comes only from the subsidy, meaning the initial price before regulation is equal to constant marginal costs (constant return to the scale). Area F represents the costs for the government for the internalization of areas B and E. The increase in the consumers' domestic surplus is still represented by area D, as in Fig 2.

Conversely, a tax could be imposed to prevent risks, $-e$. The intervention mechanisms would be alike but in the inverse direction. In a few cases, taxation systems can generate revenue to bear costs of subsidies in another product, aiming for a synergetic effect. However, identifying which products to allocate these types of interventions can be a complex task as it is hard to predict how consumers will react to price changes. Moreover, some of these interventions can cause polemical debates due to the heterogenous impact of taxation policies on consumers. For example, a per-unit subsidy(s) on a product consumed by individuals not part of the group of the population at risk would mean that $f = 0$, resulting in no returns on the costs invested for s. Likewise, a per-unit taxation could cause undesired effects such as impacting primarily on citizens who are already vulnerable due to lower wealth. In addition, some products in the market are subject to changes in their health characteristics effects, like changing their composition, e.g., fortification practices.

This theoretical section presented simplified models for a heuristic purpose. While somewhat unrealistic, we assume constant prices for the sake of exemplification of the framework. We also acknowledge that real market mechanisms are dynamic with a complex interplay of factors that are influenced by the level of information of consumers, their wealth status and market prices.

## 2.2 Case study

We investigated the impacts of a potential increase in consumption of lentils in the France based on two examples [14,52]. Our first example examines one market in relation to an increase in demand and its monetary implications in the context of microeconomics. The second example considers two markets, simulating an imperfect gram-to-gram food substitution scenario. Possibilities for further extension of the models will be later discussed.

**2.2.1 Food consumption scenarios.** We estimated the impacts of four alternative scenarios (S) when compared with the reference scenario (i.e., current baseline consumption). In S1, we investigated the impact of increasing consumption of regular consumers of lentils by 30% compared to their reference scenario. In S2, we further estimate the impacts of S1 assumptions with an additional of a 20% change of the proportion of non-consumers at the reference scenario to become lentil consumers (S1a Fig). For this, we assumed that new consumers would have similar consumption patterns to the ones in the reference scenario (i.e., mean intake and initial demand). S3 and S4 accounted for food substitutions. We formulated these scenarios based on current dietary recommendations to reduce the consumption of animal products, such as meat [8,13,53], replacing amounts by lentils. We defined meat as unprocessed red meat from any bovine (beef or veal) consumed as a steak or mincemeat. The models evaluating S3 and S4 were built on previous RBA studies [44,54] and the economic models developed by Marette et al. [50]. S3 investigated the impact of the reduction in the consumption of unprocessed red meat by 30% on a gram-to-gram substitution rate, based on cooked amounts. Because in S1-S3 we started with the health perspective (i.e., alternative scenarios defined prior to the HIA, conducting the CBA as a second step), in S4 we followed the inverse approach, allowing the results of the welfare analysis to define the proportion that would be socially acceptable to be substituted (S1b Fig).

**2.2.2 Health impact assessment.** DALYs were estimated following a top-down HIA approach [43,45], using the disease envelopes from the GBD 2019 for France [55]. The results of GBD studies including the data sources used for calculation of the GBD outputs are available in an open database [55]. Consumption data was extracted from the third French Individual and National Food Consumption Survey, INCA3 (2014–2015) [56]. The methodology applied for data collection in the INCA 3 Survey has been reported elsewhere [56]. In brief, the study followed the guidelines of the Declaration of Helsinki and was approved by the Comité Consultatif sur le Traitement de l'Information en matière de Recherche dans le domaine de la Santé (CCTIRS; Advisory Committee on Information Processing in Health Research).

 

Before enrolling in the study, verbal informal consent was obtained from all participants. Verbal consent was witnessed and recorded [56].

To filter relevant eating occasions the FoodEx2 classification standards were used (S1 Table) [57]. The scope of the health assessment was limited to adults aged ≥18 years, but data from the total population was used to inform the demand calculation. The selection of the health outcomes and data sources for computation of the DALY are detailed in S2 and S3 Tables. Uncertainty and variability around estimates were handled by applying 2-dimensional Monte Carlo simulations, drawing a set of parameters thousands of times. Statistical analyses were conducted in R version 4.2.2.

**2.2.3 Calibrating the demand.** The empirical calibration of the demand $Q(p, I_f, I_e)$ is given by Eq. (9) with unaware consumers, namely with $I_f = I_e = 0$, which are indicators equal to 1 or 0, depending on the type of information that the consumers are aware of. Using existing data on the quantity $\hat{Q}$ of the product sold over a period, the average price $p_1$ observed over the period, and the direct price elasticity of the demand $\hat{\varepsilon} = \left(\frac{dQ}{dP}\right)\left(\frac{P}{Q}\right)$ obtained from time-series econometric estimates, the calibration leads to estimated values for the demand (9) equal to $\frac{1}{\tilde{b}} = -\frac{\hat{\varepsilon}\hat{Q}}{p_1}$ and $\tilde{a} = \tilde{b}\hat{Q} + p_1$. This calibrated demand accounts for the consumers with positive purchases. For the non-purchasers (N), we assume a demand

$$QN(p, I_f, I_e) = Max\left[\frac{a' - p + I_f f - I_e e}{b}, 0\right]$$

(10)

This demand is calibrated such that $QN(p_1, 0, 0) = 0$ leading to a parameter $a' = p_1$, allowing us to capture demand changes with new consumers if the price decreases.

Results coming from a lab experiment can be integrated for determining the value of parameter $f$. These results are extrapolated to the whole population and are useful for calibrating the shifts in demand. From Eq. (3), let us consider values $WTP_1^j$ and $WTP_2^j$, where $i$ indicates participant´s WTP before and after the revelation of information. The average relative variation in WTP provides a first measure of the inverse demand shift, $\omega = \frac{[E(WTP_2) - E(WTP_1)]}{E(WTP_1)}$, where $E(.)$ denotes the expected value over all participants. A WTP gives information about the change in the inverse demand when the quantity is fixed (equal to $Q_1$ in Fig 2). The relative variation of WTP coming from the revelation of information in the lab gives an estimation of the relative variation of the inverse demand [8] related to the information with $I_f$ moving from 0 to 1. This relative variation of the inverse demand is estimated by considering the equilibrium quantity $Q_1$ on Fig 1, with $f$ defined by the following equation:

$$\omega = \frac{E(WTP_2) - E(WTP_1)}{E(WTP_1)} = \frac{p(Q_1, 1, 0) - p(Q_1, 0, 0)}{p(Q_1, 0, 0)} = \frac{f}{a - bq_1}$$

(11)

Since WTP and experimental data are subject to hypothetical bias, alternative measures can be considered for the sake of robustness, for example, the relative variation in average chosen quantities, or purchase intention (PI). This measure is represented by $\delta = \frac{[E(Q_2) - E(Q_1)]}{E(Q_1)}$, with values $E(Q_1)$ and $E(Q_2)$ indicating, for all participants, expected quantity choice of a product before and after the revelation of information (see Fig 2 with $Q_2 - Q_1 = \Delta Q$). This relative shift gives information about the shift of the demand given by [9] at the equilibrium price $p_1$, with $f$ now defined by Eq., (12).

$$\delta = \frac{E(Q_2) - E(Q_1)}{E(Q_1)} = \frac{Q(p_1, 1, 0) - Q(p_1, 0, 0)}{Q(p_1, 0, 0)} = \frac{f}{a - p_1}$$

(12)

The data necessary for computing and replicating the parameters used for the estimation of demand and calibration of economic welfare for this case study are reported in Table 2. For sake of simplicity, the parameters based on lab experiments were extracted from previous publications conducted in France [58–61]. In these past publications, PI and WTP measures were derived from anonymized answers, while informed written consent was obtained from the participants before each study took place. The studies followed the guidelines of the Declaration of Helsinki and the European General

Data Protection Regulation (GDPR) with further methodological considerations detailed elsewhere [60,61,62]. To underline the beforementioned limitations of experiments eliciting WTP, in our first example (without food substitutions) we use the parameters demonstrated in Eq. (11–12) separately, while in the second case (with food substitutions), we illustrate the impact of different sources to estimate Eq. (11) (Table 2).

For quantities consumed, we used the consumption data from INCA3, adjusting whenever possible with market data extracted from the literature (Table 2). This ensured consistency and allowed for a more detailed approach when quantifying the health impacts attributed to regular consumption habits. The use of INCA3 data for estimating demand implies that mainly the foods that are regularly consumed (i.e., frequently repeated purchases) are captured by the estimates, representing a partial representation (therefore a proxy) of the total number of yearly sales. The cross-verification of quantities consumed according to different data sources, including conversion factors applied to account for the change of weights between foods consumed and purchased (cooked versus raw), are reported in S4 and S9 Tables.

In S3 and S4, we focused on a subsegment of consumers and estimated the impact of a substitution scenario between unprocessed red meat by lentils. To calibrate these two markets, we followed the demand system with two products

**Table 2. Parameters for calibrating the demand and welfare variation used in the two examples based in the year of 2019 in France.**

| Description | Variable | Value |
|---|---|---|
| Lentil case | | |
| *From time series and observed data* | | |
| Annual consumption of lentils (kg)[a, b] | $\hat{Q}$ | 28 653 053 |
| Consumption of regular consumers (kg)[b] | $\frac{\hat{Q}}{N}$ | 0.641 |
| Average price of lentils (€/kg)[b] | $p_1$ | 2.7 |
| Own-price elasticity of demand for lentils[c] | $\hat{\varepsilon}$ | −1.23 |
| *From the lab experiment* | | |
| Relative variation in average WTP[d] | $\omega$ | 0.05 |
| Relative variation in average PI[e] | $\delta$ | 0.17 |
| Lentil versus meat case: subsegment of consumers | | |
| *From time series and observed data* | | |
| Consumption of meat (kg)[a] | $\hat{Q}_M$ | 1 078 902 205 |
| Consumption of lentils (kg)[a] | $\hat{Q}_L$ | 19 665 319 |
| Average price of lentils (€/kg)[b] | $p_1$ | 2.7 |
| Average price of meat (€/kg)[d] | $p_2$ | 10.4 |
| Own-price elasticity of demand for meat[c] | $\hat{\varepsilon}_M$ | −1.11 |
| Own-price elasticity of demand for lentils[c] | $\hat{\varepsilon}_L$ | −1.23 |
| Cross-price elasticity of demand for meat[c] | $\hat{\varepsilon}_{ML}$ | 0.02 |
| *From the lab experiment* | | |
| Relative variation in average lentils WTP[d] | $\omega$ | 0.05 |
| Relative variation in average meat WTP[d] | $\mu$ | 0 |
| Relative variation in average lentils WTP[f] | $\rho$ | 0.19 |
| Relative variation in average meat WTP[f] | $\varphi$ | −0.12 |

Note: [a] Dubuisson et al. (2019) [56], [b] Thiollet-Scholtus et al. (2024) [14], [c] for starchy foods in Caillavet et al. (2016) [58], [d] Marette (2017) [59], [e] Marette & Roosen (2022) [60], [f] Martin et al. (2021) [61].

described by Marette et al. [50]. We identified the proportion of consumers in the population which are meat consumers (57%, according to INCA 3) and further distinguished between the initial demand of a group of consumers consuming only meat (94%) and another group consuming both lentils and meat (6%). In all alternative scenarios, we calibrated the demand system of new lentil consumers assuming similar ratios and parameters determined by the initial demand of the group consuming both products. Thus, for simplification, we estimated an average price for lentils of both dry and canned products (based on conventional lentils, domestically produced) and an average price for meat regardless of the cut. Calculations were performed in Excel or Mathematica.

## 3. Results

### 3.1 Increase in consumption of lentils without food substitutions

In the reference scenario, the quantity of lentils purchased by regular consumers was 0.641 kg/person/year (Table 2), representing a mean daily intake of 2.88 g/day of cooked lentils for the total adult population, and 45.51 g/day for regular consumers only (6%). In S1 and S2, the total mean intake for the adult population would increase to 3.74 and 12.35 g/day, respectively (S5 Table).

Table 3 presents ΔDALY attributed to the consumption of lentils for S1 and S2 in relation to the reference scenario, considering the long-term protective effect of lentils consumption on the reduction of incident cases of ischemic heart disease (IHD). Detailed results of the HIA are presented in S2 Table. Results show major health gains due to a reduction in IHD (−10 619 DALYs, 95% UI: −16 765; −3 909) when a larger share of the population can benefit from the attributed health benefits linked to the consumption of lentils (S2). Table 4 shows the economic analysis for S1-S2 considering market adjustments via information campaigns with imperfect information. Furthermore, we present an additional scenario that maximizes the welfare with a per-unit subsidy (Eq. 9, Fig 3). The presented variations start from the reference scenario in which there is no policy or goal implemented.

The results presented in Table 4 suggest that changes in consumption assumed in S1 and S2 are not necessarily optimal from a social point of view. Indeed, the shifts in quantity are probably too large in these scenarios, leading to a decrease in the welfare (Fig 1). Both scenarios S1 and S2 lead to a lower welfare compared to the scenario with maximized welfare with the per-unit subsidy (s*), where an increased demand of 8–25% could be expected for up to a 3% increase in the welfare variation (1 346 603 €). Please note that revelation of perfect information would yield equivalent results as the scenario with a per-unit subsidy. In this case, the increased demand of 8–25% would be internalized by the consumers, replacing the need for a subsidy, in practice this is difficult to reach.

**Table 3. Health impact assessment: avoided new cases and averted DALYs of increasing the consumption of lentils in the French adult population for the year of 2019.**

| Health outcome | Avoided incident cases (95% UI) | ΔDALY (95%UI) |
|---|---|---|
| **Scenario S1** | | |
| IHD | −9.46 (−15.18; −3.42) | −555.16 (−901.31; −203.07) |
| **Scenario S2** | | |
| IHD | −176.77 (−276.16; −65.74) | −10 618.49 (−16 764.93; −3 909.14) |

DALY, disability-adjusted life year; UI, uncertainty intervals; IHD, ischemic heart disease.

## 3.2 Increase in consumption of lentils with decrease in consumption of meat

The mean daily intake of unprocessed red meat in the total adult population was 42.07 g/day, and 69.9 g/day for regular meat consumers only (57%). After the substitution, the meat daily intake decreased to 29.45 and 36.48 g/day for S3 and S4 (13% substitution, see S1b Fig), increasing the daily intake of lentils to 15.50 g/day and 8.47 g/day, respectively (detailed intakes in S6 Table). The DALYs attributed to the consumption of lentils and meat for the S3-S4 are presented in Table 5. To illustrate the impact of micronutrients that can also be of public health concern and that are not easily

**Table 4. Cost-benefit analysis: relative changes in absolute values (and percentages) in welfare, comparison between the alternative and reference scenarios for the year 2019.**

| Scenarios | Welfare variations with $\omega=0.05$ | Welfare variations with $\delta=0.17$ |
|---|---|---|
| **Scenario S1** | | |
| Quantity variation (kg) | 8 595 916 (+30%) | 8 595 916 (+30%) |
| Welfare variation (€) | −1 776 784 (−5%) | 361 690 (+0.9%) |
| **Scenario S2** | | |
| Quantity variation (kg) | 11 461 221 (+40%) | 11 461 221 (+40%) |
| Welfare variation (€) | −4 659 962 (−113%) | −2 609 737 (−106%) |
| **Scenario with per-unit subsidy (maximizing the welfare)** | | |
| Subsidy level (€/kg) | $s^*=0.12$ | $s^*=0.37$ |
| Quantity variation (kg) | 2 386 656 (+8.3%) | 7 273 467 (+25.4%) |
| Welfare variation (€) | 144 989 (+0.4%) | 1 346 603 (+3.2%) |

**Table 5. Health impact assessment: avoided new cases and averted DALYs in substitution scenarios of unprocessed red meat by lentils in the French adult population for the year of 2019.**

| Health outcome (food) | Avoided incident cases (95% UI) | ΔDALY (95% UI) |
|---|---|---|
| **Scenario S3** | | |
| IHD (lentil) | −4 466.72 (−7 055.09; −1 642.19) | −15 720.20 (−25 308.10; −5 779.13) |
| CRC (red meat) | −31.08 (−61.00; −1.14) | −4 668.14 (−9 371.89; −160.67) |
| T2DM (red meat) | −3 003.87 (−5 291.30; −687.35) | −419 340.45 (−2 249 151.64; −105 469.54) |
| **Total** | | −439 728.8 (−2 283 832; −111 409.3) |
| **Scenario S4[a]** | | |
| IHD (lentil) | −1 791.04 (−2 825.26; −658.30) | −6 215.01 (−9 994.13; −2 315.31) |
| CRC (red meat) | −13.94 (−27.65; −0.51) | −2 108.99 (−4 307.86; −71.15) |
| T2DM (red meat) | −1 356.21 (−2 422.36; −306) | −189 192.86 (−1 022 363.47; −46 924.73) |
| **Total** | | −195 516.90 (−1 036 665.00; −49 311.19) |

DALY, disability-adjusted life year; UI, uncertainty intervals; IHD, ischemic heart disease, CRC, colorectal cancer; T2DM, type 2 diabetes mellitus. [a]Estimates of welfare variation based on parameters from Martin et al., 2021 [61].

**Table 6. Cost-benefit analysis: relative changes in absolute values (and in percentages) in welfare for a subsegment of consumers, comparison between the alternative and reference scenarios for the year 2019.**

| Scenarios | Welfare variations, with $\omega$=0.05, $\mu$=0 | Welfare variations, with $\rho$=0.19, $\varphi = -0.12$ |
|---|---|---|
| **Scenario S3** | | |
| Quantity variation for meat (kg) | −323 670 661 (−30%) | −323 670 661 (−30%) |
| Quantity variation of lentils (kg) | 323 670 661 (1645%) | 323 670 661 (1645%) |
| Welfare variation (€) | €− 733 321 327 (−14.4%) | € −207 032 831 (−5.5%) |
| **Scenario S4 with per-unit subsidy (maximizing the welfare)** | | |
| Tax and subsidy level (€/kg) | $t^*$=0, $s^*$=0.13 | $t^*$=1.24, $s^*$=0.51 |
| Quantity variation for meat (kg) | −91 583 (−0.01%) | −144 057 789 (−13.3%) |
| Quantity variation of lentils (kg) | 21 695 019 (110%) | 83 287 707 (423%) |
| Welfare variation (€) | 1 464 413 (+0.02%) | 111 255 357 (+2.9%) |

translated into DALYs, we evaluated the impact of S3 for selected micronutrients (S2 Table) by comparing the nutrient intakes with the dietary reference values, reporting the results in S7 Table. The economic analysis for S3-S4 focusing on the subsegment of meat consumers is presented in Table 6.

Results showed that a 30% reduction in meat consumption would result in a health gain of −439 729 DALYs (95% UI: −2 283 832; −111 409) (Table 5), but this scenario would represent a change that is too drastic considering consumers preferences and welfare (Table 6). However, if fiscal interventions maximizing social welfare are implemented, almost half of the estimated health benefits in the adult population could be achieved (Table 5). These health gains are incentivized by a combination of a per-unit tax and subsidy, underlining the impact of fiscal policies for guiding the optimal substitution rate between these types of products. Implementation of a subsidy impacting the price of lentil products alone would not yield substantial differences from current consumption patterns, resulting on a substitution rate of −0.01% (welfare variation with $\omega$ and $\mu$, Table 6).

## 4. Discussion

Dietary shifts and changes in agri-food systems require public policies that address both supply and demand issues [53,63]. In this paper, we briefly reviewed current methods to evaluate the impact of interventions with different metrics to support policies for the maximization of health and social welfare and proposed a novel "expanded framework for cost–benefit analysis" that focuses on demand and fiscal instruments. Furthermore, we demonstrated the empirical application of the theoretical framework based on a relevant case study on the lentil market in France. The framework is applicable to other cases and can be extended to other parameters.

In our case study, we simulated scenarios of changes below the recommended targets of these healthy and environmentally friendly diets [8,13]. However, we observed that an increase of 30% or above in the consumption of lentils would likely not be socially acceptable, as an immediate market reaction (S1-S2). A potential increase of 8–25% in the demand for lentils could be feasible if supported by subsidies, which would decrease the price for consumers but lead to a positive welfare variation of up to a 3% increase. Similarly, despite the estimated health gains, consumers would likely be reluctant to accept a 30% substitution of lentils for unprocessed red meat (approximately −6 to −14% in welfare variation). Results of S3-S4 showed that implementing fiscal instruments that impact the market prices of both products would lead to a substitution rate that is more reasonable for these selected products from the consumer perspective (13%), resulting in −195 517 averted DALYs annually. The example also suggests that implementing subsidies for lentils only may not substantially

impact the food choices of this subsegment of consumers, leading to a small marginal increase (0.02%) in the welfare variation.

The estimates from our case study were based on relatively simple scenarios, but many extensions are possible. Both utility and demand functions were oversimplified for heuristic purposes, and more complex approaches could be envisaged for capturing the income effects and the numerous substitutions among several products. The analysis could also be refined by considering different subgroups of consumers (e.g., per sex, socioeconomic status) or even aggregated according to their reactions towards the information revealed in the lab experiments [64]. The supply side should also be integrated for considering producers and retailers. For example, extensions to increasing the supply chain (based on decreasing return to scale) are possible, which would enable us to account for profits in the supply chain. Finally, as the approach presented above captures immediate reactions to the market, discount rates reflecting repeated purchases for long-term benefits (perhaps more suitable for being compared with DALY), could also be included.

When evaluating risk-benefit management options, regulators should consider many decision factors related to (1) the effectiveness of the intervention in terms of reducing public health risk; (2) acceptability from a socioeconomic point of view; and (3) the implementation of options based on costs, feasibility, and impact [29]. In this context, our expanded CBA framework could be systematically applied to provide more concrete indications of these potential trade-offs, precising consequences in terms of DALYs, monetary values, and price distortions. This approach can provide a more quantitative evaluation than the "classical MCDA", which facilitates the ranking of most performant options based on the integration of different criteria that are either frequently expressed qualitatively or abstracted from a certain quantitative scale by the application of different value-judgment derived weights [5,29]. In addition, by interconnecting HIA and CBA in a tailored framework, we can support answering the questions: (1) "When should food regulators intervene", since both models allow for comparisons between the risks and benefits attributed to the consumption levels of a food product; and once the need for intervention is envisaged, (2) "How should food regulators intervene", as CBA is particularly suited for measuring the monetary impact of risk management options involving regulatory instruments.

Several other studies have explored the integration of epidemiological and economic models [11,36,65–68]. From the CEA perspective, Irz et al. investigated the economic feasibility of consumers adhering to nutritional recommendations by quantifying the "taste cost" of changing behaviors, i.e., the utility loss of a trade-off between a long-term health goal and a short-term pleasurable reward, by combining linear programming and epidemiological methods [36]. The framework, which was later expanded to incorporate environmental indicators [69], relied on constant prices or "shadow prices", i.e., a set of fixed prices for a nutritionally constrained individual to choose the same goods as a nonconstrained consumer. Other approaches explored the cobenefits to health, economics, and the environment of dietary shifts via a partial equilibrium model [66] but did not incorporate the evaluation of regulatory instruments. Although the applications we presented do not include environmental indicators, our models could be adapted to account for those indicators (e.g., WTP for more sustainable products). Nonetheless, the issue of incorporating the farmers´ perspective in terms of occupational health risks and mitigation strategies at the producer level from a farm-to-fork perspective remains a challenge, highlighting the need for continued research in the development of holistic approaches for food systems analysis [4].

Smith et al. have also explored ways to combine methods via QALY and WTP and developed a decision-support approach that allows the evaluation of trade-offs between health states and financial objectives while accounting for uncertainty in citizens' income [70]. These models offered the possibility for the decision-maker to adjust for different consumption patterns over time. Our models contribute differently because they provide estimates for welfare variation and evaluate the costs and impact of different regulatory tools, a need that has been neglected in previous studies [35].

Nevertheless, there are several limitations to our approach. First, we acknowledge that price is one of the key determinants of food choices and that the assumption of a constant price is unrealistic, as it can easily fluctuate depending on several market factors, including global warming and food system resilience. Second, we are aware of the overall limitations in both HIA and CBA when accounting for imperfect substitution scenarios. Although consumers' reactions to market

prices are difficult to predict, we assumed a weight-equivalent substitution rate in our health assessment, but consumers may be more concerned about isocaloric or protein-equivalent substitutions, aspects only addressed by a few RBA studies to date [71,72].

In addition, the use of population summary metrics and stated preference methods are often criticized [5,26]. Regarding HIA, one of the shortcomings of using DALY is that they capture only the health outcomes for which there is sufficient epidemiological evidence associating exposure to health outcomes and for which disability weights have been derived. As for CBA, WTP elicited by surveys or in the lab are likely to be upward biased compared to real-decision contexts, with many contributions trying to mitigate these possible biases via, for instance, follow-up questions about the certainty of responses [73]. Moreover, individuals may value short and long risks differently, over-valuing small risks while undervaluing larger ones, with a likelihood that the demand value for prevention is greater than the one for treatment [41,74].

Despite these limitations, our CBA relies on variations in WTP related to a quality shift (also called marginal WTP) via equations (3) and (11). Marginal WTP for a change in quality-related characteristic is, in general, not statistically different across hypothetical (or lab) and real payment settings [75]. Moreover, these marginal WTP tend to be stable across different experiments and contexts [76].

Another criticism of choosing CBA and WTP, and potentially RBA, is that they are resource-intensive [5,37,77]. We emphasize that this approach is suitable to inform specific decision problems in which simplified approaches are deemed inadequate. Additionally, as we demonstrated, information from the available literature can be used to inform models. To acknowledge the limitations of using metrics derived from experimental data, we presented results using different parameters and data sources, which are valuable to cross-validate and discuss outputs from this approach [61]. Furthermore, we acknowledge challenges in data alignment between the HIA and CBA. In this work, we overcame this issue by estimating demand curves based on data from the French national consumption survey, which mainly captures quantities of foods regularly consumed, i.e., repeated buyers, rather than the total amounts of products sold yearly. We took steps to quantify the impact of this decision and identify related uncertainties (S4 Table), as we recognize that INCA3 is likely outdated and no longer represents the current habits of the French population.

We encourage future applications of this framework to use more recent data and evidence, to follow best practices for conducting and reporting on HIA using RBA methods [42], and more importantly, to incorporate the model extensions presented.

## 5. Conclusions

In this work, we highlighted the gaps of current methodological approaches that support regulatory decisions and proposed a framework to more efficiently inform decisions related to fiscal interventions influencing food consumption. This interdisciplinary framework is relevant as an initiative to support dietary transitions as the WHO recommends the implementation of fiscal policies that contribute to healthy diets [1,2]. Since future challenges for achieving healthy and sustainable diets lie in a considerable change in current consumption (and therefore production) patterns, our contribution aims to support food regulators in strategizing the applicability of such diets by offering an ex ante approach to the evaluation of policies that can influence consumption habits and that is helpful in finding potential realistic and "good compromise" targets for the design of demand intervention measures.

In summary, this paper shows how the impact of regulatory options can be directly evaluated with an interdisciplinary framework, namely, by combining outputs from HIA and CBA. Both approaches can be conducted in parallel for evaluating the same regulatory decision and for obtaining a complete view of this choice. This approach can be replicated to assess the impact of regulations and be systematically implemented by other regulatory authorities around the world. The development of these complementary methods can help streamline public regulation and improve its efficiency.

 

## Supporting information

**S1 Fig. Consumption scenarios.**
(TIF)

**S1 File. Background information (HIA).**
(DOCX)

**S2 File. Supplementary information on the lentil case-study.**
(XLSX)

**S3 File. Welfare S3-S4 (CBA).**
(PDF)

## Author contributions

**Conceptualization:** Stéphan Marette.

**Formal analysis:** Constanza De Matteu Monteiro, Stéphan Marette.

**Methodology:** Constanza De Matteu Monteiro, Rodney Feliciano, Stéphan Marette.

**Project administration:** Constanza De Matteu Monteiro.

**Supervision:** Jeanne-Marie Membré, Sara Monteiro Pires, Stéphan Marette.

**Visualization:** Constanza De Matteu Monteiro, Stéphan Marette.

**Writing – original draft:** Constanza De Matteu Monteiro, Stéphan Marette.

**Writing – review & editing:** Rodney Feliciano, Jeanne-Marie Membré, Sara Monteiro Pires, Sofie Theresa Thomsen.

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
