## [Decision Letter · Decision Letter 0]

PONE-D-24-55088Disability-adjusted life years and Willingness to pay: Understanding when and how food regulators should intervene to improve consumer health through foodPLOS ONE

Dear Dr. De Matteu Monteiro,

Thank you for submitting your manuscript to PLOS ONE. After careful consideration, we feel that it has merit but does not fully meet PLOS ONE’s publication criteria as it currently stands. Therefore, we invite you to submit a revised version of the manuscript that addresses the points raised during the review process. **Both reviewers point out that the paper has some merits and deserves being considered for publication. I personally agree with their view. However, there is also scope for substantial improvements and several issues should be better clarified and discussed more in depth before the paper can be considered for publication. ** **My recommendation, when revising your paper,  is to accurately address all reviewers' concerns and to follow closely their suggestions.  **

 Please submit your revised manuscript by May 03 2025 11:59PM. If you will need more time than this to complete your revisions, please reply to this message or contact the journal office at plosone@plos.org . Please include the following items when submitting your revised manuscript:

We look forward to receiving your revised manuscript.

Kind regards,

Matteo Lippi Bruni, PhD

Academic Editor

PLOS ONE

**Journal Requirements:**

1. When submitting your revision, we need you to address these additional requirements. Please ensure that your manuscript meets PLOS ONE's style requirements, including those for file naming. The PLOS ONE style templates can be found at https://journals.plos.org/plosone/s/file?id=wjVg/PLOSOne_formatting_sample_main_body.pdf and https://journals.plos.org/plosone/s/file?id=ba62/PLOSOne_formatting_sample_title_authors_affiliations.pdf 2. Thank you for stating the following financial disclosure: Holifood project (https://holifoodproject.eu/) financed by the European Union’s Horizon programme for research and innovation (Grant Agreement No. 101059813 352).    Please state what role the funders took in the study.  If the funders had no role, please state: "The funders had no role in study design, data collection and analysis, decision to publish, or preparation of the manuscript." If this statement is not correct you must amend it as needed. Please include this amended Role of Funder statement in your cover letter; we will change the online submission form on your behalf. 3. We note that the grant information you provided in the ‘Funding Information’ and ‘Financial Disclosure’ sections do not match.  When you resubmit, please ensure that you provide the correct grant numbers for the awards you received for your study in the ‘Funding Information’ section. 4. When completing the data availability statement of the submission form, you indicated that you will make your data available on acceptance. We strongly recommend all authors decide on a data sharing plan before acceptance, as the process can be lengthy and hold up publication timelines. Please note that, though access restrictions are acceptable now, your entire data will need to be made freely accessible if your manuscript is accepted for publication. This policy applies to all data except where public deposition would breach compliance with the protocol approved by your research ethics board. If you are unable to adhere to our open data policy, please kindly revise your statement to explain your reasoning and we will seek the editor's input on an exemption. Please be assured that, once you have provided your new statement, the assessment of your exemption will not hold up the peer review process. 5. Please amend either the title on the online submission form (via Edit Submission) or the title in the manuscript so that they are identical. 6. Please include your full ethics statement in the ‘Methods’ section of your manuscript file. In your statement, please include the full name of the IRB or ethics committee who approved or waived your study, as well as whether or not you obtained informed written or verbal consent. If consent was waived for your study, please include this information in your statement as well. 7. We note that you have referenced “Thiollet-Scholtus M, Bouamra-Mechemache Z, Mouzon O de, Orozco V, Spiteri M, Membré JM” which has currently not yet been accepted for publication. Please remove this from your References and amend this to state in the body of your manuscript: (ie “Thiollet-Scholtus M, Bouamra-Mechemache Z, Mouzon O de, Orozco V, Spiteri M, Membré JM. [Submitted]”) as detailed online in our guide for authorshttp://journals.plos.org/plosone/s/submission-guidelines#loc-reference-style

Reviewers' comments:

Reviewer's Responses to Questions

**Comments to the Author**

1. Is the manuscript technically sound, and do the data support the conclusions?

Reviewer #1: Yes

Reviewer #2: Yes

2. Has the statistical analysis been performed appropriately and rigorously? 

Reviewer #1: I Don't Know

Reviewer #2: Yes

3. Have the authors made all data underlying the findings in their manuscript fully available?

Reviewer #1: Yes

Reviewer #2: Yes

4. Is the manuscript presented in an intelligible fashion and written in standard English?

Reviewer #1: Yes

Reviewer #2: Yes

5. Review Comments to the Author

**Reviewer #1: ** This is an interesting article,but the introduction is too long and difficult to read.It is difficult to understand the motivation and objective of the paper.

The authors can probably try to make a better connection between the introduction and motivation and the case study.

Why were lentils chosen for the case study? Is this an economic and consumption relevant (considerable consumption in the diet) for the area of study?

Using before and after information willingness to pay measures make exacerbate experimenter demand effects. A cleaner way to do this, particularly if doing hypothetical measures is to elicit between subject WTP and randomly assign participants to a control and an information treatment condition. Not asking you to do this, but at least acknowledge it.

Also, there are ways to mitigate hypothetical bias, by using for example donations linked to the stated WTP measures. See Elias et al (2015) as an example.

**Reviewer #2:**  The authors propose a framework for evaluating the impact of food regulations that incorporates: 1) a partial equilibrium model for measuring the welfare effect, 2) a measure of the impact of information about the food's properties on consumer preferences, and 3) the quantification of the effect in terms of health outcomes. The research is complemented by a case study on lentil consumption in France that utilizes data from the French nutritional survey (INCA3) and results from lab experiments.

It seems to me that the paper potentially contains all the elements of the story (which is very rich), but an accurate revision is needed to better clarify the overall logic and provide details to make links explicit.

For instance, although it seems a key component of the study, I'm not clear on how disclosing information about the food's nutritional properties, safety, or other attributes integrates into the framework. Is the framework intended to compare the welfare and health impacts of regulations when consumers are informed versus when they are not (so that it can be used to compare the impact of regulatory measures that are accompanied by informational campaigns with those that are not)?

The author states that the aim of the study is to develop an interdisciplinary framework to help decide “when” and “how” to implement regulatory instruments. The proposed framework seems to focus on quantifying the welfare and health income of regulatory instruments. The framework however does not directly address the temporal dimension and it is not clear to me whether it provides a guide to select among different intervention methods. In other words, it is more about measuring effects rather than providing practical guidance for choosing the timing or methods of interventions. If I have misunderstood the connection, I kindly request that it be clarified further. Otherwise, I would not emphasize the 'when' and 'how' dimension.

Perhaps the authors can add a reference for the choice of the utility function (eq(4)) which incorporates the revealed information component.

With regard to the case study, it is unclear whether an original lab experiment was conducted (lines 412–414 suggest that this is the case) or if the omega and delta parameters were obtained from other sources. If I understand correctly, the value of these parameters comes from lab experiments conducted in other studies and described in the literature. However, I believe this might be somewhat unclear to the reader, and the authors could clarify this point better for the sake of clarity. Some details about the lab experiments referenced could help complete the picture.

In line 402-403 the authors state “Since experimental data are fragile, especially because of hypothetical bias, another measure can be considered for robustness”. What is meant by fragile? Perhaps the need for an alternative measure (which would still come from experimental data) could be better argued.

In Table 2, the expressions “relative variation 1”, “relative variation 2”are, in my opinion, confusing for the reader. It is probably unnecessary to introduce numbering, and it would be sufficient to refer to “relative variation in WTP” and “relative variation in average quantities.” The same for “var1” and “var3” in Table 6.

“431 In S3 and S4, we focused on a subgroup of consumers” and Table 6 refers to “a subsegment of consumers”: Which subgroup is being referred to? Lines 434 and 435 mention two subgroups: individuals who consume only meat (94% according to INCA3?) and those who consume both meat and lentils (6% according to INCA3?). Or is it the group of meat consumers? Please, clarify.

Line 210: The sentence “The terms I_f and I_e and will equal zero when consumers are ignorant about health effects” contains an extra “and.”

Table 4 presents the impacts of scenarios 1 and 2 in terms of welfare change. The two columns show the results based on two different assumptions regarding the omega and delta parameters, which summarize the effect of the revealed information. The role of information disclosure is still unclear to me. The idea is that the regulation, which is expected to result in a 30% increase in consumption or a 20% increase in the number of individuals consuming, is accompanied by an information campaign that reveals health-related information?

In this regard, the assumption is that the change in demand due to being informed is the same for all consumers, and especially that in the case of information disclosure, all consumers are equally informed (i.e. all consumers are equally reached by the information). Perhaps this point is worth discussing more explicitly.

6. PLOS authors have the option to publish the peer review history of their article (what does this mean? ). If published, this will include your full peer review and any attached files.

**Do you want your identity to be public for this peer review?** For information about this choice, including consent withdrawal, please see our Privacy Policy .

Reviewer #1: **Yes: ** Marco A Palma

Reviewer #2: No

---

## [Author Response · Author response to Decision Letter 1]

2 May 2025

(The response to reviewers letter has been uploaded and copied here)

Dear Editor,

Thank you for the opportunity to resubmit our paper. Please find herewith the revision of our draft manuscript. We thank the two reviewers for their thoughtful comments. We did our best to incorporate all the suggestions into the revised version. We are grateful to the reviewers for their insights, and we feel that the revised manuscript significantly improved over the original version.

To address some key points made by the reviewers, we have changed the title of the manuscript, substantially restructured the introduction section and made amendments throughout the manuscript. We upload the two versions of the revised manuscript (with and without track changes) as instructed by the journal. Please note that the formatting editions done to comply with the journal requirements are not marked up. To be transparent, we include in this letter the list of references added to the revised manuscript. Furthermore, in this letter we make our response to the reviewers more explicit by highlighting in red the editions made in the text, alongside with line numbers of the revised manuscript.

Regarding the journal requirements on financial disclosure (item 2 on the PLOS ONE decision letter), we would like to declare that "The funders had no role in study design, data collection and analysis, decision to publish, or preparation of the manuscript." We thank journal for changing the online submission on our behalf.

Thanks again!

Response to Reviewer #1 Comments

Reviewer #1 comment#1 : “This is an interesting article, but the introduction is too long and difficult to read. It is difficult to understand the motivation and objective of the paper. The authors can probably try to make a better connection between the introduction and motivation and the case study. Why were lentils chosen for the case study? Is this an economic and consumption relevant (considerable consumption in the diet) for the area of study?”

*Response from the authors: Thank you for your interest and dedication in reviewing our manuscript. Thanks to your suggestions we have restructured our introduction. We hope that the revised version is easier to read, and that the objective of the paper and motive behind the choice of our case study is clarified. Since we have made considerable changes to the text, we invite the review to read again this whole section. To address the reviewer #1 specific question on why lentils were chosen, we included the paragraph below in the introduction:

“The shift to an increased consumption in legumes and other plant-based protein sources are at the core of the necessary transition towards sustainable diets (13). Lentils were deemed as a relevant case-study due to their nutritional benefits (containing protein, micronutrients, bioactive compounds), food ingredient flexibility and agricultural utility (via nitrogen fixation). Moreover, it is a widely accepted legume in France, and a crop that is expected to continue to have an increased demand in Europe (14–18).” (lines 71-77)

Reviewer #1 comment #2: “Using before and after information willingness to pay measures make exacerbate experimenter demand effects. A cleaner way to do this, particularly if doing hypothetical measures is to elicit between subject WTP and randomly assign participants to a control and an information treatment condition. Not asking you to do this, but at least acknowledge it.

Also, there are ways to mitigate hypothetical bias, by using for example donations linked to the stated WTP measures. See Elias t al (2015) as an example.”

*Response from the authors: That is a great point. We thank you the reviewer for the suggestion. Your comments helped us to reconsider this issue. We decided not to reference the study from Elias et al., (2015) as we thought that the topic the authors address in relation to donations is somehow far from the context of our study. However, we have amended the text and added other relevant references that we believe is closer to our case study context (please see the list of newly added references at the end of this letter).

While we aimed to be upfront with the limitations of WTP elicitation in the lab environment, in the revised manuscript we have tried to keep the editions concise as our paper is already quite technical and an in-depth discussion on the methodologies for derivation of WTP is not the core of our work. Therefore, in lines 607-609 we have added:

“As for CBA, WTP elicited by surveys or in the lab are likely to be upward biased compared to real-decision contexts, with many contributions trying to mitigate these possible biases via, for instance, follow-up questions about the certainty of responses (72).”

And we have emphasized on the robustness of marginal WTP for a change in quality-related characteristic in lines 613-617, which reads:

“Despite these limitations, our CBA relies on variations in WTP related to a quality shift (also called marginal WTP) via equations (3) and (11). Marginal WTP for a change in quality-related characteristic is, in general, not statistically different across hypothetical (or lab) and real payment settings (72). Moreover, these marginal WTP tend to be stable across different experiments and contexts (73).”

Please note also that “between subject WTP with randomly assign participants to a control and an information treatment condition” could be preferable but as underlined by Falk and Heckman (2009) (DOI: 10.1126/science.1168244), controlled conditions and randomized field experiments carried out under natural conditions have also limitations that may bias results.

We hope the reviewer understands our decisions in the revised manuscript.

Response to Reviewer #2 Comments

Reviewer #2 comment#1: “The authors propose a framework for evaluating the impact of food regulations that incorporates: 1) a partial equilibrium model for measuring the welfare effect, 2) a measure of the impact of information about the food's properties on consumer preferences, and 3) the quantification of the effect in terms of health outcomes. The research is complemented by a case study on lentil consumption in France that utilizes data from the French nutritional survey (INCA3) and results from lab experiments. It seems to me that the paper potentially contains all the elements of the story (which is very rich), but an accurate revision is needed to better clarify the overall logic and provide details to make links explicit.”

*Response from the authors: Thank you for your time and the thorough review of our manuscript. Thanks to your suggestion we have carefully revised our manuscript in the attempt to address your suggestions. We have changed the title of our manuscript, and we restructured the introduction considerably, including modifications in the subsection 1.1, the case-study description and discussion. We hope that the revised manuscript is more direct, making the connections more explicit.

Reviewer #2 comment #2: “For instance, although it seems a key component of the study, I'm not clear on how disclosing information about the food's nutritional properties, safety, or other attributes integrates into the framework. Is the framework intended to compare the welfare and health impacts of regulations when consumers are informed versus when they are not (so that it can be used to compare the impact of regulatory measures that are accompanied by informational campaigns with those that are not)?”

*Response from the authors: The framework is not intended to only compare the welfare and health impacts of regulations when consumers are informed versus when they are not, but also to understand: i) the current (unregulated) market effects in terms of quantification of potential externalities related to health – the costs and benefits to society, and ii) how the market could be regulated to eliminate or reduce the costs of those externalities – ways to intervene to address consumers behaviours in relation to dietary choices. In our framework we demonstrate how these externalities could be handled, i.e., internalized in the demand via information campaigns or welfare maximization, via fiscal incentives.

In other words, the framework is intended to support decision-makers on identifying the need for intervention (informed by DALY, which quantifies the health impacts) and the impacts in terms of efficacy based on monetary measures of different regulatory instruments (based on the CBA calibrated with WTP) in the context of dietary transitions. By integrating WTP in the CBA calibration we obtain more information regarding consumers preferences in relation to the market, i.e. how likely the demand could potentially change if consumers had been informed.

This helps estimating with more precision how far current habits in a population are from optimal dietary targets, and how flexible shifts in demand would be realistic considering welfare.

With this information, decision-makers can decide how to handle external costs or benefits linked to the specific food market according to (i) is it worth to intervene based on the estimated health gains, and ii) the type of intervention based on impact of different regulatory instruments. Whether the intervention should be based on policies governing awareness (i.e., information-based) or policies influencing consumers choices (i.e., incentives) will be dependent on decision-makers judgment considering their resources and priorities.

Reviewer #2 comment #3: “The author states that the aim of the study is to develop an interdisciplinary framework to help decide “when” and “how” to implement regulatory instruments. The proposed framework seems to focus on quantifying the welfare and health income of regulatory instruments. The framework however does not directly address the temporal dimension and it is not clear to me whether it provides a guide to select among different intervention methods. In other words, it is more about measuring effects rather than providing practical guidance for choosing the timing or methods of interventions. If I have misunderstood the connection, I kindly request that it be clarified further. Otherwise, I would not emphasize the 'when' and 'how' dimension.”

*Response from the authors: "We thank you for bringing up this great point. To address the reviewers comment we have changed the title and amended the text throughout the manuscript to clarify and better reflect the applicability of our framework in terms of “when” and “how”, previously stated in the original version.

We aimed at developing an improved decision-support framework by integrating in one assessment the information needed to evaluate the health and economic impacts of dietary shifts. This framework also has the asset to measure impacts of regulatory instruments, should the need for interventions be identified. We believe that the outputs provided in the framework is already valuable to inform and guide food regulators in terms of “when” and “how” to intervene. In terms of “when”, the framework provides DALY estimations with Figure 1 illustrating how different scenarios of change in consumption could impact DALY and welfare. In this figure, we suggest hypothetical areas (areas 1 to 3) for which a potential intervention would be most likely beneficial (in terms of health gains), with area 3 suggesting a potential threshold for non-intervention. As for deciding on the type of intervention, our framework is useful to understand which of the instruments would be feasible and impactful enough to reach the desirable change.

In short, the framework supports decision-making on interventions involving regulatory instruments, but it does not provide a more “in-depth practical guidance” as such, since that would require a more comprehensive overview of the needs, emergency to meet recommendations and resources available in the country, considerations and value judgments that are usually designated to the decision-maker. We briefly discuss aspects related to this point in the discussion section in lines 562-571, which reads:

“When evaluating risk-benefit management options, regulators should consider many decision factors related to (1) the effectiveness of the intervention in terms of reducing public health risk; (2) acceptability from a socioeconomic point of view; and (3) the implementation of options based on costs, feasibility, and impact (29). In this context, our expanded CBA framework could be systematically applied to provide more concrete indications of these potential trade-offs, precising consequences in terms of DALYs, monetary values, and price distortions. This approach can provide a more quantitative evaluation than the “classical MCDA”, which facilitates the ranking of most performant options based on the integration of different criteria that are either frequently expressed qualitatively or abstracted from a certain quantitative scale by the application of different value-judgment derived weights (5,29). In addition, by interconnecting HIA and CBA in a tailored framework, we can support answering the questions: (1) “When should food regulators intervene”, since both models allow for comparisons between the risks and benefits attributed to the consumption levels of a food product; and once the need for intervention is envisaged, (2) “How should food regulators intervene”, as CBA is particularly suited for measuring the monetary impact of risk management options involving regulatory instruments.”

Reviewer #2 comment#4: “Perhaps the authors can add a reference for the choice of the utility function (eq(4)) which incorporates the revealed information component.”

*Response from the authors: The text has been amended:

“The characterization of consumers’ preferences with (un)awareness regarding some characteristics follows the theorical basis of previous studies (47,48).” (lines 209-2011; see references 47 and 48)

We also acknowledge that are different approaches to integrate experimental economics than what we proposed (lines 204-205, see reference 46).

Reviewer #2 comment #5: "“With regard to the case study, it is unclear whether an original lab experiment was conducted (lines 412–414 suggest that this is the case) or if the omega and delta parameters were obtained from other sources. If I understand correctly, the value of these parameters comes from lab experiments conducted in other studies and described in the literature. However, I believe this might be somewhat unclear to the reader, and the authors could clarify this point better for the sake of clarity. Some details about the lab experiments referenced could help complete the picture.”

*Response from the authors: Indeed, the parameters were derived from the literature. We have added the following text to clarify for the readers:

“The parameters based on lab experiments were extracted from previous studies conducted in France (57-60).” (lines 427-429)

We avoided providing more detailed description of the lab experiments because our manuscript is already overall quite long.

Reviewer #2 comment#6: “In line 402-403 the authors state “Since experimental data are fragile, especially because of hypothetical bias, another measure can be considered for robustness”. What is meant by fragile? Perhaps the need for an alternative measure (which would still come from experimental data) could be better argued.”

*Response from the authors: We have replaced the word “fragile” by “limitation (or limited)” throughout the text. We tried to be upfront with the limitations in regard to WTP derived from lab experiments, reason why we have chosen to show the impact of results depending on the parameter (or parameter source).

In the response to the comments from reviewer #1

we have already made amendments to the text to better discuss the issues in relation to bias and WTP. These editions are elaborated in lines 607-609 and in lines 613-617, which reads:

“As for CBA, WTP elicited by surveys or in the lab are likely to be upward biased compared to real-decision contexts, with many contributions trying to mitigate these possible biases via, for instance, follow-up questions about the certainty of responses (72).”

“Despite thes

---

## [Decision Letter · Decision Letter 1]

Health impact assessment and cost‒benefit analysis: exploring complementarities of methods to assess the impacts of regulations on food consumption

PONE-D-24-55088R1

Dear Dr. De Matteu Monteiro,

We’re pleased to inform you that your manuscript has been judged scientifically suitable for publication and will be formally accepted for publication once it meets all outstanding technical requirements.

Kind regards,

Matteo Lippi Bruni, PhD

Academic Editor

PLOS ONE

Additional Editor Comments (optional):

Reviewers' comments:

Reviewer's Responses to Questions

**Comments to the Author**

1. If the authors have adequately addressed your comments raised in a previous round of review and you feel that this manuscript is now acceptable for publication, you may indicate that here to bypass the “Comments to the Author” section, enter your conflict of interest statement in the “Confidential to Editor” section, and submit your "Accept" recommendation.

Reviewer #1: All comments have been addressed

Reviewer #2: All comments have been addressed

2. Is the manuscript technically sound, and do the data support the conclusions?

Reviewer #1: Yes

Reviewer #2: Yes

3. Has the statistical analysis been performed appropriately and rigorously? 

Reviewer #1: Yes

Reviewer #2: Yes

4. Have the authors made all data underlying the findings in their manuscript fully available?

Reviewer #1: (No Response)

Reviewer #2: Yes

5. Is the manuscript presented in an intelligible fashion and written in standard English?

Reviewer #1: Yes

Reviewer #2: Yes

6. Review Comments to the Author

Reviewer #1: thanks for attempting to addre my suggestions. I do no have any additional suggestions after this round of reviews.

Reviewer #2: The authors have provided a thorough response to my comments. They have revised the manuscript, clarifying the motivation, structure, and application of their proposed framework. The restructuring of the introduction and the explicit discussion of the case study choice (lentils) are particularly appreciated.

Overall, the revision has improved the clarity and transparency of the paper. I recommend acceptance.

7. PLOS authors have the option to publish the peer review history of their article (what does this mean? ). If published, this will include your full peer review and any attached files.

**Do you want your identity to be public for this peer review?** For information about this choice, including consent withdrawal, please see our Privacy Policy .

Reviewer #1: No

Reviewer #2: No

---

## [Editor Report · Acceptance letter]

PONE-D-24-55088R1

PLOS ONE

Dear Dr. De Matteu Monteiro,

I'm pleased to inform you that your manuscript has been deemed suitable for publication in PLOS ONE. Congratulations! Your manuscript is now being handed over to our production team.

Kind regards,

on behalf of

Dr. Matteo Lippi Bruni

Academic Editor

PLOS ONE